# Psychometric Properties of the Intrapreneurial Self-Capital Scale in Malaysian University Students

**Chua Bee Seok [1],\*, Harris Shah Abd Hamid [2]  and Rosnah Ismail [3]**

[1]   Faculty of Psychology and Education, Universiti Malaysia Sabah, Kota Kinabalu 88400, Malaysia
[2]   Faculty of Education, Universiti Malaya, Kuala Lumpur 50603, Malaysia; harris75@um.edu.my
[3]   Department of Psychology, Faculty of Allied Health Sciences, Cyberjaya University College of Medical Sciences, Cyberjaya 63000, Malaysia; rosnahismail@gmail.com
\*   Correspondence: chuabs@ums.edu.my

**Abstract:** The Intrapreneurial Self-Capital Scale (ISCS) is a 28-item measure intended to measure individual resources used to manage career and life challenges. The Intrapreneurial Self-Capital (ISC) is a higher order construct composed of seven specific constructs: core self-evaluation, hardiness, resilience, creative self-efficacy, decisiveness, goal mastery, and vigilance. In the new research area of the psychology of sustainability and sustainable development, ISC constitutes a promising core of resources to face the challenges of the 21st century. The aim of the current study was to determine the factor structure and psychometric properties (i.e., reliability and concurrent validity) of a Malaysian version of ISCS among university students. The self-report questionnaire was administered to 1491 university students in Sabah, Malaysia. Confirmatory factor analyses were performed to assess the latent structure of the Malaysian ISCS. The final indices of Goodness of Fit showed satisfactory fit to the data. The Cronbach's alpha of the Malaysian ISCS is 0.81. The Malaysian ISCS correlates with Career Adaptability $r = 0.31$ ($p < 0.01$) and with Life Project Reflexivity $r = 0.44$ ($p < 0.01$), thus showing an adequate concurrent validity evidence. The Malaysian ISCS provides a promising research area in psychology (both positive and sustainability). Malaysian parents, teachers and counselors can also use this tool for their development and intervention efforts.

**Keywords:** intrapreneurial self-capital; factor structure; psychometric properties; reliability; concurrent validity; Malaysian version; psychology of sustainability and sustainable development

## 1. Introduction

Intrapreneurial self-capital is a new career construct proposed by Annamaria Di Fabio [1]. Intrapreneurial self-capital is the core of attributes possessed by individuals which allows them to confront life, environmental and organizational challenges, changes and transitions, and to turn the constraints into resources [1]. Individuals with high intrapreneurial self-capital are creative, innovative, careful, rational, and have higher self-determination, freedom, and autonomy [2] in problem-solving and decision-making. They plan to face uncertain environments; they study environments in depth to ensure mastery; and use information in a rational, formal, strategy-making process [1,3].

In the 21st century, Malaysia faces challenges due to globalization, internationalization and the rapid development of information and communication technology (ICT). To achieve success in the 21st century, the young generations in Malaysia are required to have intrapreneurial self-capital characteristics to face globalization and cope with the unpredictable and challenging work-life environment, and to adapt quickly to changing technologies. According to Di Fabio and Van Esbroeck [4], the intrapreneurial self-capital can be enhanced through specific training. Measurement of this construct in Malaysia is thus imperative in the effort to understanding and improving it among

the Malaysian youths. Thus, the aim of this paper is to adapt the Intrapreneurial Self-Capital Scale (ISCS) [1] by translating the items into Bahasa Melayu and validating the instrument as a bilingual questionnaire. The validated instrument then can be used to identify the necessary characteristics needed to succeed in career and life in the 21st century. Interventions to improve intrapreneurial self-capital (ISC) would also be possible with a valid measure that is locally validated.

Di Fabio [1] proposed the ISC as a higher order construct that contains seven sub-constructs: core self-evaluation, hardiness, creative self-efficacy, resilience, goal mastery, decisiveness, and vigilance. Core self-evaluation refers to individual positive judgment on their self-esteem, self-efficacy, locus of control, and absence of pessimism [5]. Hardiness (consists of three sub dimensions, commitment, control, and challenge) refers to the resistance exhibited by individuals. The sub-dimension of commitment refers to the individual tendency in engaging in all aspects of life, identifying goals, and setting priorities. The dimension of control refers to how an individual perceives their control over lives and influence the life events in a positive sense. The dimension of challenge is defined as the ability to adapt to unexpected situations, new experiences and stimulants [6]. Creative self-efficacy is referred to how an individual uses his or her ability to solve problems creatively [7]. Resilience refers to the ability to cope with and continue to withstand adversity in an adaptive way [8] and to use adaptive strategies to deal with discomfort and adversity [9]. Goal mastery is defined as one's ability to continuously develop own skills and the tendency to achieve the perceived best level in relation to each task [10]. Decisiveness is defined as one's ability to make decisions timeously in any life context [11], and refers to the characteristics relative to self-determination, freedom and autonomy in decision-making [2]. Lastly, vigilance is defined as one's ability to plan in uncertain environments, to study in-depth and to cope with the environment, and to apply the collected information in a rational, formal, and strategic way [1,2].

In the new research area of the Psychology of sustainability and sustainable development [12–14], ISC represents a promising core of resources to face the challenges of the 21st century, being associated to innovative behaviors [15] (and both hedonic and eudaimonia well-being [16,17]. ISC is an energizing and generative individual resource that could favor processes of harmonization with oneself, with others, and with nature/the natural world [18] for sustainability and achievement of sustainable development goals [19].

There are no known published studies on the translation of the ISCS into other languages. A literature search done on Google Scholar (keywords: *intrapreneurial self-capital scale* and *validation*) did not return any relevant hits. This study is the first to test the ISCS in a culturally different group compared to the ones reported in earlier studies. Thus, this paper provides an additional psychometric evidence for the ISCS beyond the original group of Italian participants on which it was tested. In the study of the original version of the ISCS [1] relationships emerged with career variables, (1) positive with career decision-making self-efficacy and perceived employability, and (2) inverse with career decision-making difficulties. ISC is configured as a career and life construct [1] and for this reason it could assume that it is positively associated with variables important for career and life construction in the 21st century. Career adaptability and life project reflexivity were thus chosen to examine the concurrent validity of the scale.

## 2. Method

### 2.1. Participants

The study participants were 1491 university students who are taking degree programs in the arts and sciences streams. There were 327 males (21.93%) and 1164 females (78.07%). The mean age was 21.67 (*SD* = 1.98). The respondents comprised 34.5% Malays, 9.1% Chinese, 2.5% Indian, 12.2% Kadazandusun, 13.2% Bajau, 2.4% Malay-Brunei, and 26.1% indicated "Other" ethnicity.

## 2.2. Measures

*Intrapreneurial Self-Capital Scale* (ISCS). The Malaysian version of the ISCS was used to evaluate ISC. This Malaysian version was obtained through back-to-back translation method of the English version supplied by the author. The tool consists of 28 items with a response format on a five-point Likert scale from 1 = *Strongly agree* to 5 = *Strongly disagree* as used in the original instrument. Examples of items are: "Planning in advance can help avoid most future problems"; "I feel I'm able to produce innovative ideas"; "One of my goals is to acquire new skills". The Cronbach's alpha coefficient for the original Italian version was 0.84 [1].

*Career Adapt-Abilities Scale* (CAAS). The CAAS [20] is composed of 24 items with a response format on a five-point Likert scale from 1 = *Not Strong* to 5 = *Strongest.* The scale has four dimensions: Concern (example of item: "Thinking about what my future will be like"; Control (example of item: "Taking responsibility for my actions") Curiosity (example of item: "Looking for opportunities to grow as a person); Confidence (example of item: "Taking care to do things well"). The Cronbach's alphas in the present study were: 0.91 for Concern, 0.91 for Control, 0.92 for Curiosity, 0.92 for Confidence, 0.97 for the total score. The scale was back-translated by the same persons doing the back-translation of the ICS.

*Life Project Reflexivity Scale* (LPRS). The LPRS [21] is composed of 15-items with a response format on a five-point Likert scale from 1 = *strongly disagree* to 5 = *strongly agree.* The scale has three dimensions: Clarity/Projectuality (example of item: "The projects for my future life are clearly defined"); Authenticity (example of item: "The projects for my future life are full of meaning for me"); Acquiescence (example of item "The projects for my future life are more anchored by the values of the society in which I live than my most authentic values"). The Cronbach's alphas coefficient in the present study were: 0.87 for Clarity/Projectuality, 0.88 for Authenticity, 0.83 for Acquiescence 0.90 for the total score. The same back-translation procedure mentioned for the two earlier scales was followed for this scale.

## 2.3. Procedure

Data were collected from undergraduate students from one of the public universities in Kota Kinabalu, Sabah. The students were in their second and third year of studies. Respondents of the current study were recruited for the online study using the university's online platform regularly used by the students. The information related to the study and the questionnaire was given to the potential respondents by the lecturers and with the help of the students. The respondents involved in this study were those who had responded to the online study.

The original ISCS is in Italian and the first three authors had done the back-to-back translation into Bahasa Malaysia via an English version supplied by original instrument's author. Considering the relative novelty of the instruments, and the mixed language competencies of the target respondents, the instruments were presented to the respondents as a bilingual questionnaire because Bahasa Malaysia is an official language used in Malaysia and English is a second language that widely understood.

## 2.4. Data Analysis

The factorial structure of the Malaysian ISCS was tested using Confirmatory Factor Analysis (CFA) with AMOS (maximum likelihood method). The confirmatory factor analysis was carried out to confirm the higher order structure with the nine first order constructs scores as indicators as for the Italian version. The first order constructs are the following: (1) core self-evaluation, (2) commitment dimension of hardiness, (3) control dimension of hardiness, (4) challenge dimension of hardiness, (5) creative self-efficacy, (6) resilience, (7) goal mastery, (8) decisiveness, and (9) vigilance.

Different indices were used to estimate the fit of empirical data to the theoretical model: the Tucker-Lewis Index (TLI), the Comparative Fit Index (CFI), the Root Mean Square Error of

Approximation (RMSEA) and the Standardized Root Mean Squared Residual (SRMR). For the TLI [22,23] and CFI [22] values of 0.90 and higher are indicators of a good fit. Values of the RMSEA and SRMR less than 0.08 indicate a good fit [24,25]. The reliability of the Malaysian ISCS was verified using Cronbach's alpha coefficient. Concurrent validity was examined through correlations of Malaysian ISCS with Career Adaptability Scale and Life Project Reflexivity Scale. Hierarchical regressions were also carried out with control variables (gender, age) in the first step and ISCS in the second, and CAAS and LPRS as outcomes.

## 3. Results

Confirmatory Factor Analysis confirmed the higher order structure with the nine first order constructs scores as indicators for the Italian version. On the basis of modification indices analysis, an opportunity emerges to add some co-variances. The following four co-variances between the errors were inserted: error of decisiveness with error of core self-evaluation; error of creative self-efficacy with error of resilience; error of goal mastery with error of hardiness commitment; error of hardiness control with error of hardiness commitment. The final indices of *Goodness of Fit* are reported in Table 1 and they are satisfactory. The model itself is presented in Figure 1.

**Table 1.** Confirmatory Factor Analysis: *Goodness of Fit* (*N* = 1491).

|      | TLI  | CFI  | RMSEA            | SRMR |
|------|------|------|------------------|------|
| ISCS | 0.91 | 0.94 | 0.06 [0.06–0.07] | 0.04 |

**Figure 1.** Structural model of the Malaysian Intrapreneurial Self-Capital Scale (ISCS).

The Cronbach's alpha for the Malaysian ISCS is 0.81. Total-item correlations are reported in Table 2.

**Table 2.** Total-item correlations.

| Item | Total-item Correlation |
|------|------------------------|
| ISCS1 | 0.20 |
| ISCS2 | 0.20 |
| ISCS3 | 0.33 |
| ISCS4 | 0.36 |
| ISCS5 | 0.35 |
| ISCS6 | 0.34 |
| ISCS7 | 0.28 |
| ISCS8 | 0.38 |
| ISCS9 | 0.39 |
| ISCS10 | 0.39 |
| ISCS11 | 0.19 |
| ISCS12 | 0.12 |
| ISCS13 | 0.06 |
| ISCS14 | 0.47 |
| ISCS15 | 0.50 |
| ISCS16 | 0.42 |
| ISCS17 | 0.46 |
| ISCS18 | 0.38 |
| ISCS19 | 0.33 |
| ISCS20 | 0.47 |
| ISCS21 | 0.47 |
| ISCS22 | 0.42 |
| ISCS23 | 0.27 |
| ISCS24 | 0.08 |
| ISCS25 | 0.23 |
| ISCS26 | 0.44 |
| ISCS27 | 0.45 |
| ISCS28 | 0.40 |

For the concurrent validity, the Malaysian ISCS correlated in the expected direction with Career Adaptability $r = 0.31$ ($p < 0.01$) and with Life Project Reflexivity $r = 0.44$ ($p < 0.01$). These correlations showed an adequate concurrent validity of the scale. The full correlation matrix among the studied variables with also means and standard deviations is reported in Table 3.

**Table 3.** Means, standard deviations, and correlations of the studied variables.

| | *M* | *SD* | ISC | CAAS | LPRS |
|------|------|------|------|------|------|
| ISCS | 98.98 | 9.90 | - | | |
| CAAS | 83.08 | 20.71 | 0.31 ** | - | |
| LPRS | 54.44 | 8.09 | 0.44 ** | 0.21 ** | - |

$N = 1491.$ ** $p < 0.01.$

The results of hierarchical regressions with control variables (gender, age) in the first step and ISC in the second, and CAAS and LPRS as outcomes are reported in Table 4. The results show that CAAS and LPRS explain additional variances of the ISCS and both of them are significant. In other words, the construct measured by ISCS is explained better by CAAS and LPRS after controlling for gender and age, thus providing an additional evidence of the ISCS construct validity.

**Table 4.** Hierarchical regressions.

|  | CAAS | LPRS |
|---|---|---|
|  | β | β |
| *Step 1* |  |  |
| Gender | 0.07 | 0.02 |
| Age | 0.01 | 0.04 |
| *Step 2* |  |  |
| ISC | 0.31 *** | 0.44 *** |
| $R^2$ *step 1* | 0.00 | 0.00 |
| $\Delta R^2$ *step 2* | 0.10 *** | 0.20 *** |
| $R^2$ *total* | 0.10 *** | 0.20 *** |
|  | (1) |  |

$N = 1491$. *** $p < 0.001$.

## 4. Discussion

This study was carried out to adapt the ISCS as a bilingual questionnaire called the Malaysian ISCS. Data from undergraduate students show that the validity of the Malaysian ISCS is supported by evidence from CFA (the data fits the original factor structure) and external validation (correlation with similar constructs and regressions). The internal consistency is also satisfactory. Although the ISCS factorial structure shows satisfactory fit indices for the final model, the challenge and decisiveness dimensions, even if statistically significant, appear less robust in our sample (as can be seen from the factor saturations in Figure 1 and total-item correlations in Table 2). Future research should focus on assessing the full cross-cultural invariance of ISC construct. Nevertheless, reliability and concurrent validity measures of ISC, suggested that ISCS could be used effectively for career construction and primary preventive perspective also in Malaysia, for building strengths for the 21st century challenges, according to psychology of sustainability and sustainable development.

These findings are encouraging for future investigation into intrapreneurial self-capital construct in the Malaysian context. Based on the confirmation of a single higher order structure, a single overall score of the ISCS is recommended for use in research and practices focusing on intrapreneurial self-capital. On the other hand, further psychometric investigations should examine the individual factors of ISCS given that the ISC is a core of intrapreneurial resources. The resources includes the positive self-evaluation of the self-concept [5], the hardiness exhibited by individuals as commitment, control and challenge [6], the self-efficacy to solve creatively problems [7], the strengths to transform constraints into resources [8,9], the pursuit to develop one's own skills [10], the perceived ability to decide regarding every aspect of life [11], and to make decisions carefully and adaptively [2]. The ISC provides a core of personal strengths to deal with the constant changes, transitions and challenges of career and life [1].

Future research could examine the distribution of the ISCS scores in the wider population. Given that male respondents comprise less than 22% of the participants, the findings from this study might be influenced more by female respondents. While the proportion itself is not surprising given the prevailing male-to-female ratio in universities in Malaysia, the true psychometric properties of the Malaysian ISCS might have been biased. The gender differences (e.g., similarity of scores, measurement invariance) were not highlighted in previous studies (e.g., [1,17]). However, the attention to gender differences might be warranted. A meta-analysis done on another measure of personal resources (Self-Compassion Scale) found that males have higher scores than females, albeit with small effect size [26]. Therefore, it would be worth investigating if the same pattern holds true for ISCS to provide further convergent evidence for the validity of ISCS in measuring personal resources.

The bilingual questionnaire was used to maximize comprehension by the respondents. The approach seems to be successful given the high internal consistencies of the scales. Future studies should consider a monolingual questionnaire. Validation of the Bahasa Melayu version of

ISCS on its own is desirable to ascertain its utility amongst Malay-speaking population. From a test development perspective, a monolingual questionnaire would add incremental validity evidence for the instrument beyond the adapted bilingual version as used in this study. In fact, further translation and validation of the ISCS are necessary to provide more comprehensive psychometric evidence.

The possibility of having the ISCS available also in the Malaysian context could open future perspectives for research and intervention in the new research area of psychology of sustainability and sustainable development [12–14] also in a cross-cultural perspective. In terms of research, the ISCS can extend the current investigations into personal resources like grit [27], self-compassion [28], mattering [29], and developmental assets [30] by focusing on the role of those resources directly for career and life challenges. These constructs go beyond what was already mentioned in the Introduction section. For example, further construct validation can be made among ISCS and those constructs. Another line of possible research is testing the career-specific orientation of the ISCS against the more generic personal resources. Given the size of the data set available, a norm for the ISCS scores could be derived. However, it is beyond the scope of this paper to be presented here. The current data set can be complemented with more diverse groups (especially in terms of age and educational level) from the Malaysian population to get a comprehensive norm.

With cumulative evidence on the psychometric soundness of the Malaysian (and the stand-alone Bahasa Melayu version of ISCS), it would be a very useful tool for parents, counselors, teachers and other stakeholders who are interested in helping adolescents and young people. For example, 90% of school dropout cases were attributed to lack of interest among students in one of the state in Malaysia [31]. In tackling this issue, the Malaysian ISCS is useful in determining the level of core strengths necessary to do well in school and to keep the will to be in school. The counselors can communicate the results of the assessment to parents and teachers to better coordinate the efforts to reduce school dropout cases.

This example of coordinated efforts would be achievable using locally-validated tests for assessing personal strength in the context of career and life challenges. However, such tests are sparse in Malaysia. For example, the Self-Compassion scale that was translated into Bahasa Melayu [28] has undergone limited validation. On the other hand, the study on Grit [27] uses the original English items. Thus, this paper offers the practitioners and stakeholders a tool with better psychometric properties.

**Author Contributions:** Conceptualization, R.I.; Methodology, C.B.S.; Software, C.B.S.; Validation, C.B.S., and H.S.A.H.; Formal Analysis, C.B.S.; Investigation, all the authors; Resources, all the authors; DataCuration, all the authors; Writing-Original Draft Preparation, all the authors; Writing-Review & Editing, C.B.S. and H.S.A.H.; Visualization, C.B.S.; Supervision, R.I.; Project Administration, C.B.S.; Funding Acquisition, R.I.

**Funding:** This research received no external funding.

**Conflicts of Interest:** The authors declare no conflict of interest.

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
