# Peer review of "Psychometric Properties of the Intrapreneurial Self-Capital Scale in Malaysian University Students"

_sustainability, doi:10.3390/su11030881_

Round 1
Reviewer 1 Report
The paper “Psychometric Propertises of the Intrapreneurial Self-Capital Scale in Malaysian University Students” investigates the properties of a 28-item scale that measures 7 dimensions of Intrapreneurial Self-Capital, considering a wide sample of Malaysian students. The study is interesting and has potential useful implications for the context. Nevertheless, some minor revisions and integrations are needed in order to make the manuscript suitable for publication in Sustainability.
Title
Change Propertises with Properties.
Abstract
- I suggest restructuring the abstract, since currently it is more focused on the introduction of the topic and study’s results, and misses conclusions and considerations about practical implications of the study.
Introduction
- Check the use of abbreviations. Generally, the first time the word should be used in full, followed by the abbreviation in parentheses. Thereafter use only abbreviation.
- According to the title, which refers to testing psychometric properties of the scale, the study means an adaptation, instead of validation (as you mentioned at rows 57 e 58).
- The introduction misses references to previous studies’ results on Intrapreneurial Self-Capital and its link with career outcomes. In particular, authors should argument why they have chosen some specific variables in order to test concurrent validity.
Method
Participants
- Since the study involved a very wide sample of university students, it would be useful to report what they studied.
- Moreover, why did author decide to specify participants’ religion?
- Finally, Row 103 misses an information after 10.8%.
Measures
- The ISCS’s Likert scale, where 1 is strongly agree and 5 is strongly disagree, is opposite compared to the other scales. Could authors explain why and how they managed it?
- Row 133: change “for .87” in “.87 for”
- Row 148: considering that three authors proposed the paper, who the “fourth author” is?
Results
This section needs significant improvement, since the main results are not clearly presented. In particular, this section misses:
- descriptive statistics (mean and standard deviation) of each measure.
- correlations table, including all factors, other measured variables and demographic variables (e.g. age, gender, …)
- CFA results, since fit indices are not enough. A figure or table with all paths (factor loadings, covariances, errors, …) is fundamental to understand and evaluate the final model.
- if authors tested alternative factor models they should present and compared fit results.
- total-item correlation
Moreover, presenting items in English and Malaysian versions would benefit readers and future scales’ users.
Discussion
In the discussion section, authors should discuss practical implications of the study, describing how the instrument may be used and linkde to other interventions in this field.
Author Response
We thank both reviewers for their constructive feedback in improving the manuscripts. Below are the summary of feedback and changes done to the manuscripts as suggested by both reviewers.
Reviewer 1
Comments and Suggestions for Authors | Remarks and feedback |
Change Propertises with Properties.
| Changed |
Abstract | |
- I suggest restructuring the abstract, since currently it is more focused on the introduction of the topic and study’s results, and misses conclusions and considerations about practical implications of the study. | The previous abstract was shortened and statements regarding practical implications were added. |
Introduction | |
- Check the use of abbreviations. Generally, the first time the word should be used in full, followed by the abbreviation in parentheses. Thereafter use only abbreviation. | The abbreviations have been revised for consistency of use. |
- According to the title, which refers to testing psychometric properties of the scale, the study means an adaptation, instead of validation (as you mentioned at rows 57 e 58). | The description of the purpose of the study was changed to include adaptation. The same point is also changed in the Discussion section. |
- The introduction misses references to previous studies’ results on Intrapreneurial Self-Capital and its link with career outcomes. In particular, authors should argument why they have chosen some specific variables in order to test concurrent validity. | We argumented why we have chosen some specific variables in order to test concurrent validity. |
Method | |
Participants | |
- Since the study involved a very wide sample of university students, it would be useful to report what they studied. | Added – the students were from art and science stream. |
- Moreover, why did author decide to specify participants’ religion? | Deleted. |
- Finally, Row 103 misses an information after 10.8%. | Added – Kadazandusun |
Measures | |
- The ISCS’s Likert scale, where 1 is strongly agree and 5 is strongly disagree, is opposite compared to the other scales. Could authors explain why and how they managed it? | The original instrument uses the stated scale. A statement is added to emphasise this point. |
- Row 133: change “for .87” in “.87 for” | Changed |
- Row 148: considering that three authors proposed the paper, who the “fourth author” is? | This was a mistake. The statement has been changed. |
Results | |
This section needs significant improvement, since the main results are not clearly presented. In particular, this section misses: | |
- descriptive statistics (mean and standard deviation) of each measure. | Descriptive statistics (mean and standard deviation) of each measure are reported in Table 3. |
- correlations table, including all factors, other measured variables and demographic variables (e.g. age, gender, …) | As suggested also by the second review the full Pearson's correlation matrix with all the measures included in the study, with APA format (Means, SD, and correlations) is included in the article. |
- CFA results, since fit indices are not enough. A figure or table with all paths (factor loadings, covariances, errors, …) is fundamental to understand and evaluate the final model. | We provided in Figure 1 the path diagram for the final model in which factor loadings and covariances are explicit. Further, we discussed in the discussion section the critical issues related to the challenge and decisiveness dimensions (Line 274-280). |
- if authors tested alternative factor models they should present and compared fit results. | We did not test alternative factor model because we would like to replicate the structure of the original Italian version |
- total-item correlation | We reported in Table 2 the total-item correlations. |
Moreover, presenting items in English and Malaysian versions would benefit readers and future scales’ users. | The items in English are available under request from the author of original version of the ISCS: Annamaria Di Fabio, University of Florence, Florence, Italy. Items in Malaysian version are available under request from the author: Chua Bee Seok, Universiti Malaysia Sabah, Sabah, Malaysia. |
Discussion | |
In the discussion section, authors should discuss practical implications of the study, describing how the instrument may be used and linked to other interventions in this field. | Three paragraphs were added to include the suggested elaboration. |
Reviewer 2 Report
Thank you for giving me the opportunity of revising your paper. I think that your manuscript can be improved in some ways before being published.
Please, expand your discussion on the constructs that ISC includes using the current literature.
Please, better justify your decision of taking CAAS and Life-project reflexibility in order to test the validity of ISC.
I strongly recommend you to include the full Pearson's correlation matrix with all the measures included in the study, with APA format (Means, SD, and correlations). The absence precludes future inclusion of your paper in any meta-analysis.
Perhaps, you can strength your paper by including a regression analysis with control variables (gender, age, educational level, socioeconomic position) in the first step, and ISC in the second, and CAAS or Life-project as outcomes.
Finally, your paper has a lot of implications for the further development of research, but also for parents, teachers, and counselors. Please, expand them.
Author Response
We thank both reviewers for their constructive feedback in improving the manuscripts. Below are the summary of feedback and changes done to the manuscripts as suggested by both reviewers.
Reviewer 2
Comments and Suggestions for Authors | Remarks and feedback |
Please, expand your discussion on the constructs that ISC includes using the current literature. | We expanded the discussion adding reference to the constructs included in the ISC. |
Please, better justify your decision of taking CAAS and Life-project reflexibility in order to test the validity of ISC. | We justified our decision to use CAAS and LPR in order to test concurrent validity of the ISC. |
I strongly recommend you to include the full Pearson's correlation matrix with all the measures included in the study, with APA format (Means, SD, and correlations). The absence precludes future inclusion of your paper in any meta-analysis. | The full Pearson's correlation matrix with all the measures included in the study, with APA format (Means, SD, and correlations) is included in the article. |
Perhaps, you can strength your paper by including a regression analysis with control variables (gender, age, educational level, socioeconomic position) in the first step, and ISC in the second, and CAAS or Life-project as outcomes. | We report a regression table with available control variables (gender, age) in the first step and ISC in the second, and CAAS or Life-project as outcomes. |
Finally, your paper has a lot of implications for the further development of research, but also for parents, teachers, and counselors. Please, expand them. | Three paragraphs were added to include the suggested elaboration. |